# Cardiac Magnetic Resonance Imaging in the Evaluation of Functional Impairments in the Right Heart

**DOI:** 10.3390/diagnostics14222581

**Published:** 2024-11-17

**Authors:** Andra Negru, Bogdan M. Tarcău, Lucia Agoston-Coldea

**Affiliations:** 1Department of Internal Medicine, Iuliu Hatieganu University of Medicine and Pharmacy, 400347 Cluj-Napoca, Romania; luciacoldea@yahoo.com; 2Doctoral School of Biomedical Science, University of Oradea, 1 University Street, 410087 Oradea, Romania; tarcaubogdanmihai@gmail.com

**Keywords:** cardiac magnetic resonance imaging, right heart, right myocardial strain imaging, biomechanics

## Abstract

Cardiac magnetic resonance (cMRI) imaging has recently become essential in cardiology. cMRI is widely recognized as the most reliable imaging technique for assessing the size and performance of the right ventricle. It allows for objective and functional cardiac tissue evaluations. Early in disease progression, cardiac structure and activity decrease subclinically. Late-phase clinically visible signs have been associated with less favourable outcomes. Subclinical alterations ought to be recognized for rapid evaluations and accurate treatment. An increasing amount of evidence supports cMRI deformation parameter quantification. Strain imaging enables cardiologists to assess heart function beyond traditional measurements. Prognostic information for cardiovascular disease patients is obtained through the right ventricle (RV) strain, including information primarily about the left ventricle (LV). Right atrial (RA) function evaluations using RA strain have been promising in recent studies. Therefore, this narrative review aims to present an overview of the data that are currently available for assessing right myocardial strain and biomechanics using cMRI.

## 1. Introduction

Biomechanics regulate the heart’s function in both standard and pathological conditions. Mechanical stress and biochemical mechanisms regulate myocyte activity and the extracellular matrix’s structure, affecting the heart’s performance in healthy and diseased conditions. Myocytes shorten the collagen matrix, creating forces for chamber emptying and filling. When the heart deforms, the active and passive components of the myocardium collaborate to preserve the overall muscle tension, alignment, ventricular size, and shape [1]. Failure to adapt to homeometrics leads to enlarged sizes, affecting diastolic ventricular interactions, limiting exercise capacity, and causing vascular congestion [2]. Speckle tracking echocardiography (STE) and cardiac magnetic resonance imaging (cMRI) can assess right ventricular (RV) strain. However, its complex structure and overload response make it challenging to detect dysfunction using two-dimensional STE. Three-dimensional STE offers vector data but requires multiple software packages [3]. Cardiac magnetic resonance imaging (cMRI), the gold standard for chamber function assessments, offers a quick and accurate method for measuring RV sizes, shapes, and functions. It uses four-dimensional acquisition, three-dimensional data, and postprocessing software, reducing the time and effort associated with an analysis. Its high temporal and spatial resolutions ensure reliable measurements [4].

This narrative review aims to present an overview of the currently available data on assessing right myocardial strain and biomechanics using cMRI.

## 2. RV Mechanical Pattern

Three main mechanisms operate the RV pump: the stretching of the interventricular septum and its expansion into the RV, causing anteroposterior shortening; the internal progress of the RV free wall (radial shortening); and longitudinal shortening with tricuspid valve traction towards the apex [5]. The tricuspid annulus’s traction towards the apex causes longitudinal shortening. In contrast, the free wall’s radial motion—often called the “bellows effect”—and the chamber’s anteroposterior shortening—when the free wall is stretched over the septum—all contribute to the RV’s complex contraction pattern [6]. In healthy circumstances, longitudinal shortening had been believed to account for most RV pump activity, but new investigations show that both longitudinal and radial movements are equally significant [7].

Myocardial strain is a dimensionless parameter determined by the length difference between two locations prior to and following motion. Strain echocardiography is a noninvasive method used to monitor RV systolic function, assess myocardial contractility, and identify subclinical myocardial alterations. By analyzing the many ways in which the myocardium changes shape, the following are described: longitudinal strain (RVGLS), which is the length decrease from base to apex; radial strain (RVGRS), which refers to the myocardial deformation that occurs in a radial direction towards the middle of the chamber; and circumferential strain (RVGCS), which can be determined by measuring the myocardial fibre contraction around the RV’s circumference in a short-axis view. Its accuracy, particularly in RVGLS, makes it a valuable tool for early subclinical dysfunction detection [8]. The strain rate quantifies the deformation speed [9]. An RV strain analysis uses the RVGLS total and RV free wall longitudinal strain (RVFWLS).

## 3. cMRI Techniques Used to Assess RV Myocardial Strain

cMRI is presently considered the most reliable and widely accepted method for assessing RV function and volume [2]. A recent study revealed that RVGLS provides a more accurate predictive indicator than conventional metrics. RVGLS is a significant and independent prognostic indicator of adverse outcomes in dilated cardiomyopathy (DCM), hypertrophic cardiomyopathy (HCM), arrhythmogenic RV cardiomyopathy (ARVC), and amyloidosis [10].

Myocardial strain measurements are achievable using cMRI tagging. Initially, strain imaging was performed using a traditional tagged cMRI approach. Feature tracking imaging (FTI) quantifies strain using cine pictures of the cardiac cycle [11]. FTI has been the most widely used strain cMRI technique, even though the endocardial and epicardial boundaries are the primary areas of interest. Two-dimensional (2D) cMRI-FTI strain analysis is widely recognized as an effective method for identifying early anomalies in the ventricular myocardium. Two-dimensional cMRI-FTI is an established technique that calculates strain values by separately analyzing short-axis and long-axis cine images [12]. Due to a homogeneous mid-myocardial texture without a target feature, it is challenging to measure strain in the mid-myocardial layer using cMRI-FTI [13]. So far, some studies have suggested that a three-dimensional (3D) strain cMRI-FTI analysis could address the geometry-dependent limitations of 2D strain analyses by utilizing the intrinsic directions of deformation. Consequently, many 3D feature tracking technologies are already accessible, while their expertise remains constrained. When this extension is viable, local 3D tissue characteristics may be monitored concurrently in all directions to ascertain all deformation parameters. This could diminish deformation artefacts, such as those arising from the through-plane displacements of three-dimensional structures [14].

Strain-encoded (SENC) cMRI, a recently developed tagged technology, outperformed the cMRI-FTI approach in terms of performance. The SENC technique incorporates through-plane tags to quantify strain in a direction that is perpendicular to the imaging plane [15]. Utilizing cMRI-based tissue tracking (cMRI-TT) allows for the retrospective investigation of dynamic deformation. Conversely, cMRI tagging can be utilized to investigate the process of ventricular untwisting [16].

Tissue displacement is converted into the image phase using displacement encoding with stimulated echoes (DENSE). DENSE-quantitative cMRI detects myocardial strain and displacement with superior spatial and temporal sensitivities. A quick cine DENSE pulse sequence was established by echo combination reconstruction with intrinsic phase correction and balanced steady-state free precession (b-SSFP). This sequence produces good-quality strain with an effective breath hold [17].

## 4. cMRI RV Strain in Normal Individuals

The literature has documented the establishing of reference ranges for left LV strain and strain rates through cMRI-TT techniques. The availability of data that concern the right heart, such as cMRI-TT, still needs to be improved in terms of their clinical significance [18]. In a study conducted by Truong et al., it was demonstrated that an association exists between gender and the peak RVGLS in a cohort of 50 healthy volunteers. The RVGLS was measured to be −22.11% ± 3.51% [19]. In a study by Liu et al., the authors examined the typical characteristics of the RV long axis in a sample of 100 healthy volunteers. The results indicated that the RVGLS was measured at −24.2% ± 3.59%, while the peak systolic strain rate was determined to be −1.54 ± 0.41. Nevertheless, the study did not consider the impact of the cardiac strain rate and age variations [20]. In a study performed by Liu et al., which attempted to examine the normal reference values of RV myocardial strain parameters utilizing cMRI-TT technology in a larger cohort of Chinese individuals, the RVGLS and RVGRS values were between −24.3 ± 4.7 and 23.0 ± 8.5 [18]. Strain measurements obtained via FTI have been demonstrated to be comparatively lower in patients and healthy individuals, in contrast to the strain measurements obtained using the SENC technique. Moreover, the magnitude of strain values exhibited a more comprehensive range when employing the FTI rather than the SENC technique. Table 1 shows further normal RV strain values obtained using cMRI and method variations [21,22,23,24], and Figure 1 shows normal RV strain.

**Table 1 diagnostics-14-02581-t001:** RV myocardial strain variations in normal individuals detected by cMRI.

First Author	Year	N	Findings
Barison et al. [21]	2023	100 healthy subjects	RVGLS different between genders: (total): −23.9 ± 4.1 (m) −23.0 ± 3.6 vs. (f) −24.7 ± 4.3 RVGLS values showed a weak connection with body surface area and no correlation with age or heart rate RVGLS was associated with RV end-diastolic volume
Yang et al. [22]	2023	44 studies with 3359 healthy subjects	RVGLS: −24.0%
Li et al. [23]	2022	566 healthy Chinese subjects	Women presented greater values of RVGRS than men RVGRS: (f) 25.1 ± 7.8% vs. (m) 22.1 ± 6.7% RVGCS: (f) −14.4 ± 3.6% vs. (m) −13.2 ± 3.2% RVGLS: (f) −22.4 ± 5.2% vs. (m) −20.2 ± 4.6% Ageing was linked to higher RVGRS, RVGCS, and RVGLS
Voges et al. [24]	2022	157 healthy pediatric subjects	RVGLS −24.9 ± 4.3 in patients older than 10 years (m) −24.1 ± 4.1 vs. (f) −26.2 ± 4.2 Age, height, and weight affected RVGLS exclusively in males

Abbreviations: f, females; m, males; N, number of patients; RVGCS, RV global circumferential strain; RVGLS, RV global longitudinal strain; RVGRS, RV global radial strain.

## 5. cMRI RV Strain in Cardiovascular Diseases

### 5.1. Pulmonary Artery Hypertension

The SENC cMRI method showed lower RVGLS and RVGCS values in pulmonary arterial hypertension (PAH) patients compared to healthy controls [25]. A possible noninvasive alternative to the existing invasive techniques for assessing RV–arterial coupling and end-diastolic stiffness in patients with PAH is cMRI-determined RV strain, which is linked with RV–arterial uncoupling and RV end-diastolic stiffness in chronic RV overload [26]. In a study conducted by Cao et al., which included a cohort of 79 individuals consisting of 47 patients diagnosed with PAH and 32 healthy individuals, the participants received cMRI utilizing an SSFP sequence in order to assess the function of both their LV and RV. The RVGRS and RVGCS of patients with PAH presented a statistically significant decrease compared to those in the healthy control group. The RVGRS and RVGCS had a moderate association in the group PAH. The group with an RV ejection fraction (RVEF) less than 40% had a notably lower absolute RVGCS of −7.71 ± 2.90%, as well as a lower apical RVGCS of −6.38 ± 2.62% and a mid-segment RVGCS of −8.05 ± 4.40%, in comparison to both the group with an RVEF of more than or equal to 40% and the healthy control group. However, no significant difference was observed in PAH patients with reduced RVEF [27]. RA strain and RVGLS predict major adverse cardiac events (MACEs) independently of clinical and imaging factors. The incremental utility of RA and RV strain tests in detecting RV function changes even when the RVEF is unaltered could potentially guide prognosis in precapillary PAH (pPAH) patients [28]. RV strain can identify preclinical PAH and evaluate illnesses and prognoses [29]. The pulmonary artery can be seen in Figure 2 using a short-axis cine cMRI.

Recently, three-dimensional (3D) phase-contrast magnetic resonance imaging, known as 4D flow cMRI, has been established as a minimally invasive technique for quantitatively assessing the blood flow through the heart and major arteries during cardiac cycles. Additionally, the distinctive blood flow seen in pulmonary hypertension may be shown using 4D flow cMRI. This technique may serve as a valuable, noninvasive, nonionizing approach for diagnosing and monitoring patients with PAH [30]. Numerous quantitative cMRI parameters have been documented to exhibit significant alterations in pulmonary hypertension, facilitating the precise assessment of cardiac function and morphology. Notably, the emergence of the vortical blood flow in the main pulmonary artery has been evaluated using 4D flow imaging. Furthermore, the length of the vortical blood flow was proposed as a viable 4D flow measure for the precise noninvasive assessment of PAH, making cMRI 4D flow imaging very compelling for noninvasive PAH monitoring [31]. Pulmonary artery hypertension observed using 4D flow can be seen in Figure 2.

### 5.2. Heart Disease

In patients with a corrected tetralogy of Fallot (TOF), axial slices were more repeatable for measuring the RV volume than short-axis slices. Late gadolinium enhancement (LGE) images improved the contrast to reveal ventricular fibrosis information related to clinical arrhythmia and exercise intolerance [4]. Research on children who had a TOF surgically repaired found an inverse relationship between the RV strain and strain rate measurements from echocardiography and the amount of pulmonary regurgitation as measured by cMRI [32]. Children with an RV regional dysfunction might be identified using corrected RVEF based on a net-forward pulmonary flow [33]. For individuals whose TOF has been surgically corrected, strain cMRI was considered more practical in evaluating RV strain [34]. cMRI helps determine surgical efficacy and challenges in TOF congenital heart disease. Moon et al. employed cMRI-FTI to demonstrate that TOF patients had significantly reduced RV strain values comparted to normal patients and that RVGLS was strongly connected with poor outcomes. In many corrected TOF patients, RVLS and RVCS were related to their NYHA class. RVGLS predicted adverse arrhythmic outcomes [35,36]. A cMRI-FTI strain study empirically assessed global/regional RV dysfunction and dyssynchrony in ARVC patients, facilitating early diagnosis [37]. TOF using cMRI can be seen in Figure 3.

A myocardial deformation analysis using cMRI-derived strain is an increasingly used method for evaluating ventricular function in patients receiving Fontan palliation. Two-dimensional cMRI feature tracking techniques that characterize RVGCS or RVGLS provide superior spatial resolution without constraints from acoustic windows. This presents a significant benefit for individuals with an intricate ventricular morphology undergoing surgical palliation, leading to myocardial scarring and dyssynchrony. Research has more often favoured the use of strain-based measures to identify modest subclinical alterations in ventricular contractile function [38]. Myocardial segment-specific strain–time curves characterize classic-pattern dyssynchrony and have recently been applied to patients after Fontan procedures [39]. In individuals with a single right ventricle morphology, myocardial strain patterns undergo dynamic changes during and after single-ventricle palliation. In individuals with hypoplastic left heart syndrome and its pathological variations, the RVGCS/RVGLS ratio may approach 1.0 and may completely revert in some patients long after their Fontan procedure [40].

The transposition of the great arteries after an atrial switch surgery and the congenitally corrected transposition of the great arteries are often linked to compromised systemic RV function and a poor prognosis. Diller et al. demonstrated that global longitudinal systolic strain is significantly diminished in patients with a systemic RV, correlates with subpulmonary ventricular function, and predicts adverse clinical outcomes in adults with an atrial switch transposition of their great arteries [41]. Samarai et al. have shown that in patients with a congenitally repaired transposition of their great arteries or those who have had Mustard or Senning operations, RVGLS may be beneficial for assessing systemic RV function [42].

### 5.3. Arrhythmogenic Right Ventricular Cardiomyopathy

ARVC can be diagnosed by the presence of ventricular dysfunction and ventricular arrhythmias (VAs). Compared to RV kinetic abnormalities, fat infiltration lacks specificity and is less repeatable than most cMRI abnormalities [43]. cMRI-FTI successfully and reproducibly assessed regional strain in the RV myocardium in instances of ARVC. Vigneault asserts that the results of their study indicate that cMRI FTI has the potential to increase the accuracy of ARVC diagnosis and the detection of irregularities in regional wall motion [44]. In a study conducted by Bourfiss et al., it has been demonstrated that patients with ARVC who have a sustained VA over their follow-up period exhibit a decrease in both their RVGLS and RVGCS [45]. Numerous research studies have been conducted to ascertain the presence of an abnormal RV strain and strain rate, including in the RV free wall and sub-tricuspid area, in individuals with ARVC. There is a correlation between abnormal strain and strain rate in the RV and the evolution of ARVC. Moreover, the phenomenon of mechanical dispersion, which refers to the standard deviation of the time it takes for the strain to reach its peak in a model of the RV divided into multiple segments, was observed to be more prominent in patients with early-stage ARVC compared to patients with ventricular tachycardia originating from their RV outflow tract. Additionally, it was discovered that mechanical dispersion serves as an indicator of previous arrhythmia events in patients with early-stage ARVC. The integration of data from mechanical dispersion and deformation patterns into RV analyses has led to a gradual enhancement in the identification of individuals who have had life-threatening VA in its early stages [46]. A cMRI-FTI strain study empirically assessed global/regional RV dysfunction and dyssynchrony in ARVC patients, facilitating early diagnosis [37]. The appearance of ARVC using different cMRI techniques can be seen in Figure 4.

### 5.4. RV Infarction

LGE is typically used to diagnose RV infarctions on cMRI in individuals with acute inferior infarction, as seen in Figure 5. In a study by Kumar et al., all individuals with LGE RV also had aberrant RV wall motion. On the other hand, nine individuals without LGE RV also had aberrant RV wall motion. At 13 months of follow-up, the LGE evidence suggested irreversible RV damage in the acute phase [47]. The identification of the pattern of LGE’s presentation is essential. LGE in the ischaemic pattern consistently affects the subendocardial layer, exhibiting varying degrees of transmural involvement. It is confluent and localized within the territory of a single coronary artery. In a non-ischaemic pattern, LGE does not meet the established criteria, presenting as midwall, subepicardial, or mixed, and it is not necessarily confluent or restricted to the territory of a single coronary artery. Using the secondary eigenvector from cardiac diffusion tensor imaging, the assessment of sheetlet orientation changes correlates with impaired radial strain. Segments exhibiting reduced subendocardial cardiomyocytes, indicated by a diminished proportion of myocytes displaying a right-handed orientation on helix angle maps, have compromised longitudinal strain. Infarct segment enhancement shows a significant correlation with the secondary eigenvector and right-handed orientation. The data indicate a correlation between the myocardial microstructure and contractility post-myocardial infarction, implying a possible clinical significance of cMRI cardiac diffusion tensor imaging in terms of functional outcomes [48].

Furthermore, hepatic T1 mapping has emerged as a potential imaging biomarker for the cardio–hepatic axis in ST-elevation myocardial infarction, correlating with right ventricular involvement and elevated NT-proBNP levels [49]. The RVFWLS parameter exhibited the ability to differentiate between acute ischemia and non-ischemic myocardium. Furthermore, the RVGLS had value as an independent predictor of functional status [50]. In myocardial infarction, the right ventricle functions as a significant prognostic indicator for forecasting long-term outcomes, exceeding the importance of myocardial viability [51].

### 5.5. Hypertrophic Cardiomyopathy

cMRI-derived RV global strains and various regional longitudinal strains offer distinct radiological information for differentiating between amyloidosis and HCM [52]. Right ventricular global strains correlate with outcomes in patients with HCM, including those without RV hypertrophy. RVGLS and RVGCS serve as independent predictors of HCM in cases with and without RV hypertrophy [53]. Despite a normal LVEF, Mahmod et al. demonstrated RV and LV strain impairments in HCM patients. The RVEF decreased slightly but was within normal limits [54]. Although Mahmoud et al. found no association between the RVGLS decline and composite cardiovascular events, they found it did predict non-sustained ventricular tachycardia. Hence, the independent predictive significance of RV strain must be carefully examined in relation to conventional cardiac function [54]. RV long-axis strain (RV-LAS) is the length change from the LV apex to the tricuspid annulus. Hence, it may include both ventricles’ longitudinal functions. Yang et al. observed that RV-LAS predicted the poor prognoses of HCM patients [55].

### 5.6. Amyloidosis

Right ventricular strain assessed via cMRI also serves as a significant prognostic marker for mortality in cardiac amyloidosis [56]. cMRI tissue tracking is a practical and reproducible RV deformation analysis method that could assist AL-type amyloidosis patients in detecting RV involvement early. AL-type amyloidosis mortality can be predicted by RVGLS [57]. Yang et al. demonstrated, in a cMRI-FTI study on over 300 patients, that mixing a biventricular strain analysis with an RVEF evaluation enhanced the detection of adverse outcomes: MACEs were predicted most accurately by RVGRS. A cMRI-FTI deformation analysis of both ventricles has a prognostic effectiveness comparable to traditional techniques [58].

### 5.7. Dilated Cardiomyopathy

Recent studies indicate that RVGLS assessed through cMRI plays a significant role in reclassifying the risk of major cardiovascular events in patients with DCM [59]. The predictive value of the peak of RVGLS derived from cMRI-FTI in DCM patients with stage C or D heart failure (HF) and no atrial fibrillation (AF) has been documented by Liu et al. [60]. RV-LAS showcased the highest level of diagnostic accuracy in DCM patients, according to Arenja et al. Additionally, according to their research, RV-LAS was an independent MACE marker in a multivariate analysis and essential in prognosis predictions [61].

## 6. Right Atrial Strain

For reliable RA function measurements, the RA strain appears promising. STE and, more recently, cMRI-FTI reveal reservoir, conduit, and booster atrial activity [62]. The three primary phases of the right atrium’s contraction can be seen in Table 2. RA dyssynchrony can also be determined using the reservoir or atrial contractile phases’ time to peak strain. Like STE, cMRI-FTI uses b-SSFP cine images to calculate quantitative deformation parameters without additional tagging sequences. cMRI provides benefits, although more research is required on RA strain assessments using cMRI versus STE [63]. Maceira et al. applied the short-axis method to obtain a reference range for all RA reservoir, conduit, and booster pump parameters, dividing patients into subjects, males and females, age groups, and BSA-normalized values. The complete group had a total emptying fraction of 56  ±  7.0%, a passive emptying fraction of 36  ±  7.3%, and an active atrial emptying fraction of 30  ±  8.3% [64].

Li et al. investigated 624 patients, and, after 32.5 months, 205 (32.9%) met the composite HF objective. Compared to healthy people, DCM patients had reduced RA function. Their RA reservoir strain and conduit strain independently predicted all-cause mortality. Additionally, RA strain outperformed baseline clinical and cMRI indicators in predicting MACEs [65]. Vos et al. examined RA and RV function changes in pPAH patients utilizing FTI myocardial deformation. It was proven that pPAH patients had a lower RVEF; longer RV contraction durations; and an impaired RA strain, RVGLS, and RVGCS compared to healthy controls. The RA and RV strain changed even in pPAH patients with a maintained RVEF. This group of pPH patients had a reduced conduit strain and RA reservoir. Their RVGLS was damaged, but RVGCS remained unharmed. This increased their RVCS/RVLS ratio, making their RV function more dependent on pPAH circumferential shortening. RVGLS and all RA phasic strain factors independently predicted MACEs. Research by Kutty et al. found abnormalities in the RA global longitudinal strain (RAGLS), RA end-diastolic volume index, and RA ejection fraction in 171 TOF patients and 140 healthy controls. A reduced RAGLS indicates a lower RA reservoir function [28]. Table 3 summarizes various cMRI- and STE-based studies on RA myocardial strain [66,67,68,69,70,71,72], which are also seen in Figure 6.

**Table 2 diagnostics-14-02581-t002:** The right atrium’s three primary phases.

Primary Phases	Physiological Function
The reservoir phase	This functions as storage for systemic blood following the closure of the tricuspid valve. This phase is contingent upon the RV’s longitudinal contraction, the RA’s compliance, and the return of the caval venous blood flow. Elevation during this phase contributes to enhanced RV filling [73].
The conduit phase	This facilitates the passive filling of the vena cava following the opening of the tricuspid valve. This phase relies on ageing and RV diastolic function [73].
The active contraction phase	This phase is characterized by the presence of a sinus rhythm and functions as a “booster pump” for end-diastolic atrial contraction, thereby facilitating RV filling. This phase relies on the contractile properties of the atrial wall and ventricular end-diastolic pressure [73].

**Table 3 diagnostics-14-02581-t003:** Various cMRI- and STE-based studies on RA myocardial strain.

First Author	Year	n	Disease	Values	Findings
Schönbauer et. al. [66]	2023	188	AF versus sinus rhythm	Reservoir strain, % 28 ± 12 Conduit strain, % 15 ± 9 Booster strain, % 13 ± 5 Reservoir strain rate, %/s 134 ± 63 Conduit strain rate, %/s 115 ± 67 Booster strain rate, %/s 148 ± 54	Impaired RA reservoir strain, strain rate, conduit strain, and conduit strain rate significantly correlated with worse outcomes for sinus rhythm patients.
Cengiz Elçioğlu et. al. [67]	2023	54	Isolated Atrial Septal Aneurysm (ASA)	RARS (RA reservoir strain) in ASA vs. normal individuals 36.97 ± 2.19 vs. 39.77 ± 2.36%, RAPCS (RA peak contraction strain) in ASA vs. normal individuals 16.78 ± 2.10 vs. 18.54 ± 2.43%	ASA patients had substantially reduced RARS and RAPCS. Strong independent correlation between ASA, RARS, and RAPCS.
Nyberg et al. [68]	2023	1329	Healthy individuals	RARS 38.1 (17.2–58.9) RACDS (RA conduit-phase strain) –21.0 (–37.7 to –4.2) RACTS (RA contraction-phase strain –17.1 (–28.2 to –6.0)	RACTS contractile strain values increased with age.
Emre et al. [69]	2023	175	Osteoponegrin and right heart function in hypertensive individuals with normal EF	RARS (%) 40.5 ± 11.8 RACDS (%) –22.18 ± 7.8 RACTS (%) –18.5 ± 7.7	Significant differences were observed in strain during RARS and RACDS. RACDS was identified as an independent predictor of elevated osteopontin.
Tomaselli et. al. [70]	2023	132	RARS is helpful for predicting AF recurrence following electrical cardioversion.	Recurring AF patients have lower RARS than SR individuals: 14% ± 10% vs. 20% ± 9%	After electrical cardioversion, the RARS was independent and AF recurrence was highly predicted.
Hinojar et. al. [71]	2023	176	Severe TR	RARS 11.2 (7–16) RACDS −11.2 ± 6 RACTS −2.5 ± 7	At least severe TR patients had reduced RARS and RACDS compared to controls and AF patients. Patients having events had lower RARS and RACDS levels. RARS was more significantly linked to outcomes than RACDS.
Coskun et al. [72]	2023	101	ASDs	Before and six months after the treatment, the mean RA appendage and global strain was −13.31 ± 4.84% and −18.53 ± 4.69%, respectively	RA global strain and appendage strain can improve following transcatheter ASD closure.

Abbreviations: AF, atrial fibrillation; ASA, Isolated Atrial Septal Aneurysm; ASDs, Ostium secundum; EF, ejection fraction; f, females; n, number of patients; m, males; RA, right atrium; RAPCS, RA peak contraction strain; RARS, RA reservoir strain; RACDS, RA conduit phase strain; RACTS, RA contraction phase strain; TR, tricuspid regurgitation.

## 7. Conclusions

Table 4 summarizes the parameters of right ventricular function derived from cardiac magnetic resonance imaging and details the clinical contexts in which they have proven useful.

**Table 4 diagnostics-14-02581-t004:** Parameters of right ventricular function derived from cardiac magnetic resonance imaging and their clinical function.

Cardiac Magnetic Resonance Imaging Right Ventricular Parameter	Clinical Function
RVGLS	Independent prognostic factor for adverse outcomes in DCM, HCM, ARVC, and AL-type amyloidosis [10]. RVGLS predicts MACEs independently of clinical and imaging factors in PAH patients [28]. TOF patients had significantly reduced RVGLS, and it was strongly connected with poor outcomes. RVGLS predicted adverse arrhythmic outcomes in TOF patients [35]. RVGLS may be beneficial for assessing systemic RV function in the transposition of the great arteries or those who have had Mustard or Senning operations [42]. Patients with ARVC who have sustained VA exhibit a decrease in RVGLS [45]. The RVGLS has value as an independent predictor of functional status in patients with RV infarction [50].
RVGRS	The RVGRS of patients with PAH presented a statistically significant decrease compared to that of healthy individuals [27]. MACEs are predicted most accurately by RVGRS in cardiac amyloidosis [57].
RVGCS	The RVGCS of patients with PAH presented a statistically significant decrease compared to that of healthy individuals [27]. Patients with ARVC who have sustained VA exhibit a decrease in RVGCS [45].
RVFWLS	The RVFWLS parameter exhibited the capability to differentiate between acute ischemia and non-ischemic myocardium in patients with RV infarction [50].
Right atrial strain	pPAH patients had an impaired RA strain compared to healthy controls [28]. The RA changed even in pPAH with a maintained RVEF. This group of pPH patients had a reduced conduit strain and RA reservoir [28]. All RA phasic strain factors independently predicted MACEs in PAH patients [28]. Reduced RAGLS indicates lower RA reservoir function in TOF patients [28]. RA reservoir strain and conduit strain independently predicted all-cause mortality in DCM patients [60]. RA strain outperformed baseline clinical and cMRI indicators in predicting MACEs in DCM patients [65].

Abbreviations: ARVC, arrhythmogenic right ventricular cardiomyopathy; DCM, dilated cardiomyopathy; HCM, hypertrophic cardiomyopathy; RA, right atrium; RVGCS, RV global circumferential strain; RVGLS RV global longitudinal strain; RVGRS, RV global radial strain; RVFWLS, RV free wall longitudinal strain RVFWLS; RV-LAS, the length change from the LV apex to the tricuspid annulus; RAGLS, RA global longitudinal strain; RVEF, RV ejection fraction; pPAH, precapillary PAH, i.e., pulmonary arterial hypertension; MACEs, major adverse cardiac events; TOF, tetralogy of Fallot; VAs, ventricular arrhythmias.

## 8. Typical Pitfalls When Using Strain Imaging

The primary limitation of utilizing strain imaging is attributed to inter-observer bias. Although numerous software companies have attempted to automate analyses, the majority remain only semi-automated and frequently require manual adjustments to contours, thereby introducing the potential for human error. A study by the EACVI/ASE Industry Task Force reported a mean relative intra-observer difference in GLS of 6.1% and an inter-observer difference of 7.0%. Another significant limitation is the absolute difference in GLS among various vendors. The EACVI/ASE Industry Task Force undertook an Inter-Vendor Comparison, revealing a variance of 3% in the mean GLS across various vendor software platforms [74]. Exam times are also increasing as additional metrics are introduced to the procedure and 3D/4D becomes more widespread. The slow adoption of strain imaging can largely be attributed to its cost. Although many vendors provide strain modules tailored to their machines, the associated costs can be quite high, making obtaining strain analyses on more affordable or older equipment challenging. Also, accurate strain measurements rely heavily on training and experience [75].

## 9. Future Directions

Machine learning has been used in cMRI centres to measure cardiac mass and function, achieving results equivalent to or better than human experts, but with limited sample numbers [76]. Tan et al. investigated how a convolutional network, a deep learning method, could automatically segment the LV into short-axis slices. Tan et al. applied a machine learning technique to public datasets, including the LV segmentation challenge dataset, demonstrating its potential in automated LV segmentation using cMRI [77]. Bai et al. calculated LV and RV masses and volumes using a fully convolutional network to analyze 93,500 images from 5000 individuals [78]. Machine learning is crucial in medical applications, wearables, and miniaturized devices, but its challenges include large datasets, tedious data exchanges across institutions, and complex institutional review boards. Innovative software could help academic centres integrate clinical and imaging data using machine learning [79].

## Figures and Tables

**Figure 1 diagnostics-14-02581-f001:**
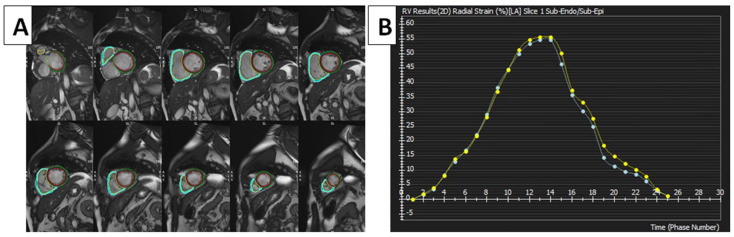
FTI strain measurements of RV using cMRI tissue-tracking software (cvi42 version 5, Circle Cardiovascular Imaging Inc., Calgary, AB, Canada). (**A**) FTI with radial strain overlaid on short-axis cine SSFP image. Green circle: epicardium of LV; red circle: endocardium of LV; blue circle: epicardium of RV; yellow circle: endocardium of RV. (**B**) RV global radial strain. Radial strain values are usually positive. On the graph, the vertical axis shows the radial strain and the horizontal axis shows the time in milliseconds. FTI, feature tracking imaging; RV, right ventricle; LV, left ventricle; cMRI, cardiac magnetic resonance imaging; SSFP, steady-state free precession.

**Figure 2 diagnostics-14-02581-f002:**
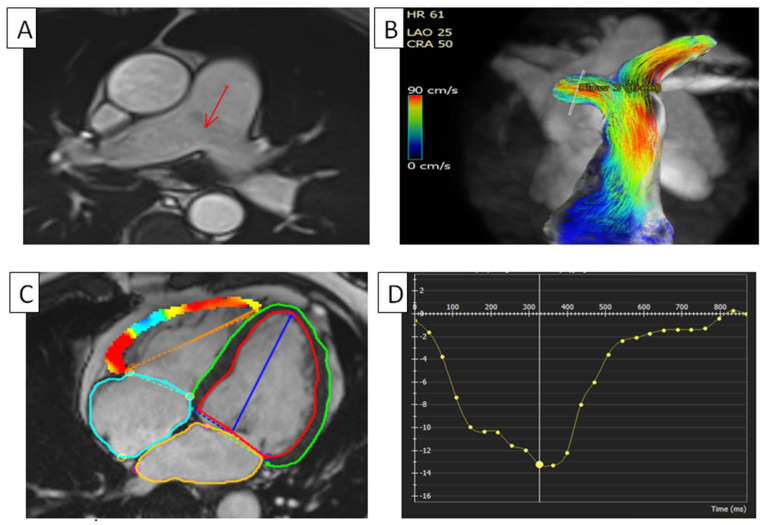
RV in pulmonary hypertension. (**A**) Dilated pulmonary artery in can be seen in a patient with pulmonary hypertension using short-axis cine cMRI. The arrow is pointing to the dilated pulmonary artery. (**B**) Pulmonary artery flow, using cMRI phase contrast, can be seen in a patient with pulmonary hypertension. (**C**) Four-dimensional myocardial FTI on a short-axis cine SSFP image. (**D**) RV global longitudinal strain in a patient with pulmonary hypertension. Longitudinal values are usually in the negative range. On the graph, the vertical axis shows the longitudinal strain (14.1%) and the horizontal axis shows the time in milliseconds. FTI, feature tracking imaging; RV, right ventricle; cMRI, cardiac magnetic resonance imaging; SSFP, steady-state free precession.

**Figure 3 diagnostics-14-02581-f003:**
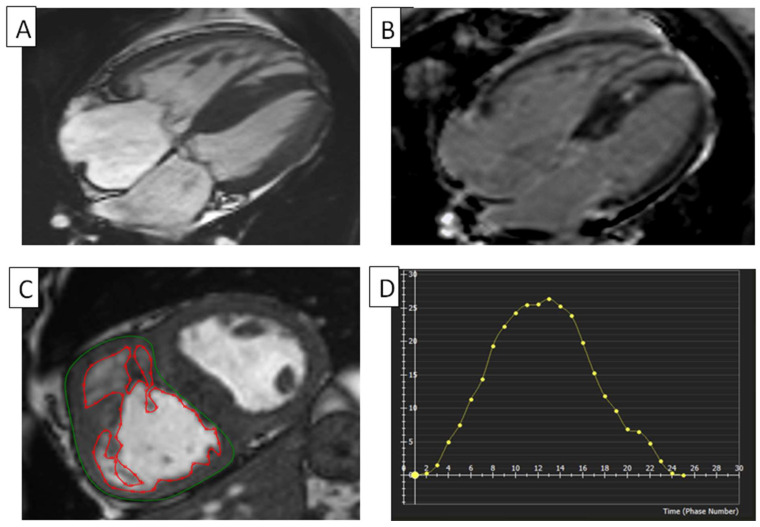
RV in a tetralogy of Fallot. (**A**) Cine cMRI four-chamber view of a patient with TOF. The RV is mildly dilated and has mild RV hypertrophy. The LV has severe hypertrophy. (**B**) cMRI, with phase-sensitive inversion recovery and four-chamber view, of a patient with TOF, showing a dilated RV and RA. (**C**) Planimetry of RV. The RV myocardium’s epicardial (green) and endocardial (red) borders are manually traced at the end-diastole on this short-axis cMRI in a patient with TOF showing increased trabeculation in their right ventricle. (**D**) RV global radial strain in TOF. Radial strain values are usually positive. On the graph, the vertical axis shows the radial strain and the horizontal axis shows the time in milliseconds. RV, right ventricle; LV, left ventricle; cMRI, cardiac magnetic resonance imaging; TOF, tetralogy of Fallot.

**Figure 4 diagnostics-14-02581-f004:**
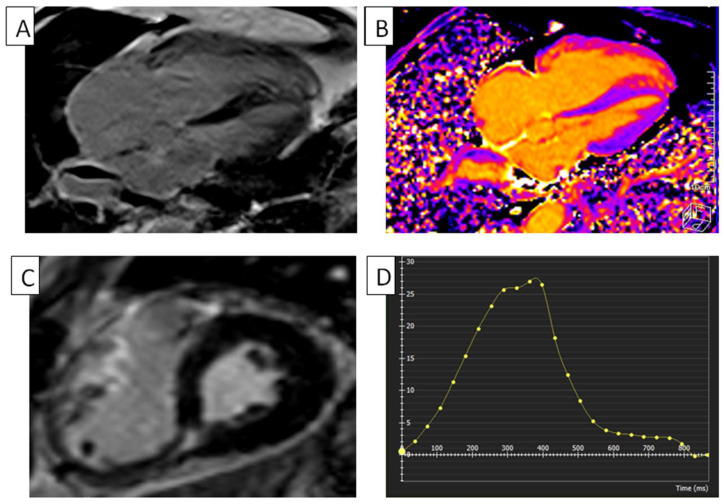
RV in ARVC. (**A**) cMRI, with phase-sensitive inversion recovery and four-chamber view, of a patient with ARVC, showing severe dilated and hyper trabecular RV and dilated RA. (**B**) T1 mapping of a five-chamber view of a patient with ARVC, used to assess myocardial extracellular space, showing severe RV hyper trabeculations and dilatation. (**C**) Short-axis phase-sensitive in-version recovery cMRI view of a patient with ARVC, showing severe RV dilatation. (**D**) RV global radial strain in a patient with ARVC. Radial strain values are usually positive. On the graph, the vertical axis shows the radial strain and the horizontal axis shows the time in milliseconds. ARVC, arrhythmogenic right ventricular cardiomyopathy; RA, right atrium; RV, right ventricle; cMRI, cardiac magnetic resonance imaging.

**Figure 5 diagnostics-14-02581-f005:**
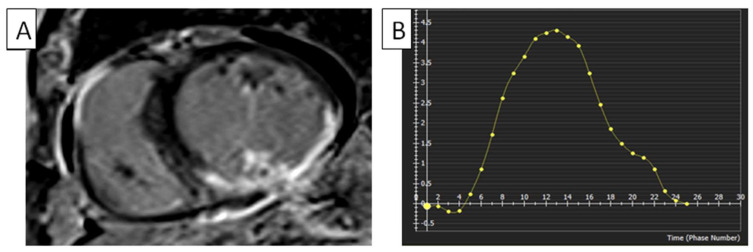
RV infarction. (**A**) Short-axis phase-sensitive inversion recovery, late-phase contrast enhancement cMRI view of a patient with inferolateral LV and RV lateral wall infarction. (**B**) Two-dimensional myocardial FTI on a short-axis cine SSFP image shows an impaired RV radial strain. Radial strain values are also usually in the positive range. On the graph, the vertical axis shows the radial strain and the horizontal axis shows the time in milliseconds. FTI, feature tracking imaging; LV, left ventricle; RV, right ventricle; cMRI, cardiac magnetic resonance imaging; SSFP, steady-state free precession.

**Figure 6 diagnostics-14-02581-f006:**
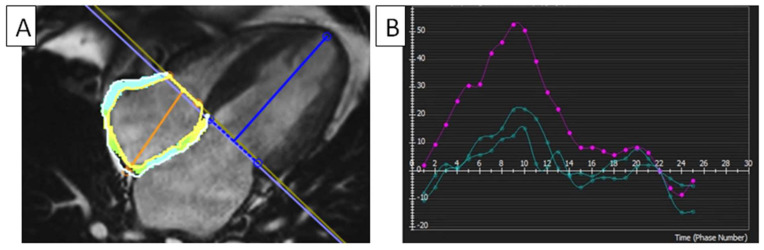
Right atrial strain. (**A**) RA strain analyzed using cMRI-FTI and a four-chamber cine SSFP image in healthy men. (**B**) RA radial time–strain curves (purple) measured using feature tracking cMRI show that the RAGLS’s reservoir function is 54%, its conduit is 17.3%, and its boost is 15.7%. FTI, feature tracking imaging; RA, right atrial; RV, right ventricle; RAGLS, right atrial global longitudinal strain; cMRI, cardiac magnetic resonance imaging; SSFP, steady-state free precession.

## Data Availability

The data presented in this study are available upon request from the corresponding author and are subject to personal data protection regulations.

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
