# Peer review of "Cardiac Magnetic Resonance Imaging in the Evaluation of Functional Impairments in the Right Heart"

_diagnostics, 2024, doi:10.3390/diagnostics14222581_

Round 1
Reviewer 1 Report
Comments and Suggestions for Authors
The authors provide an overview of the usefulness of cardiovascular magnetic resonance strain measurements to assess the performance of the right ventricle. The manuscript is informative and provides new information. I have a few comments:
Comments:
1) "cMRI techniques used to assess RV myocardial strain": Although 2D-CMR-FT is commonly used, I would appreciate if you could explain 2D and 3D CMR feature tracking.
2) In the chapter “cMRI RV strain in cardiovascular diseases – pulmonary arterial hypertension” a sentence about 4D Flow is provided without further explanation (Pulmonary artery hypertension using 4D flow can be seen 159 in Figure 2.). Please give further information or delete this part.
3) Congenital heart disease: only information about tetralogy of Fallot is provided. I suggests including studies about patients in Fontan circulation or patients with transposition of the great arteries and suggest to reduce the part about tetralogy of Fallot.
4) I think that the manuscript would benefit from some general images about Feature tracking and the parameters that we get for RV and RA assessment.
5) In addition, a table that explains the measurements (e.g. atrial reservoir, conduit, and booster function) would also be helpful.
6) SENC / DENSE is better explained than routine FT but only feature tracking images are shown. Maybe this part in the text can be reduced.
7) I recommend including a chapter about typical pitfalls (maybe before future directions).
Author Response
1) "cMRI techniques used to assess RV myocardial strain": Although 2D-CMR-FT is commonly used, I would appreciate if you could explain 2D and 3D CMR feature tracking
Thank you for pointing this out. We agree with this comment. Therefore, we have updated the text:
Two-dimensional (2D) cMRI-FTI strain analysis is widely recognized as an effective method for identifying early anomalies in the ventricular myocardium. 2D cMRI-FTI is an established technique that calculates strain values by separately analyzing short-axis and long-axis cine images [12]. Due to a homogeneous mid-myocardial texture without a target feature, it is challenging to measure strain in the mid-myocardial layer by cMRI-FTI [13]. So far, some studies have suggested that three-dimensional (3D) strain cMRI-FTI analysis can address the geometry-dependent limitations of 2D strain analysis by utilizing the intrinsic directions of deformation. Consequently, many 3D feature tracking technologies are already accessible, while their expertise remains constrained. When this extension is viable, local 3D tissue characteristics may be monitored concurrently in all directions to ascertain all deformation parameters. This could diminish deformation artefacts, such as those arising from through-plane displacements of three-dimensional structures [14].
2) In the chapter “cMRI RV strain in cardiovascular diseases – pulmonary arterial hypertension” a sentence about 4D Flow is provided without further explanation (Pulmonary artery hypertension using 4D flow can be seen 159 in Figure 2.). Please give further information or delete this part.
Thank you for pointing this out. We agree with this comment. Therefore, we have updated the text:
Recently, three-dimensional (3D) phase-contrast magnetic resonance imaging, known as 4D flow cMRI, has been established as a minimally invasive technique for quantitatively assessing blood flow through the heart and major arteries during cardiac cycles. Additionally, the distinctive blood flow in pulmonary hypertension may be shown using 4D flow cMRI. This technique may serve as a valuable, noninvasive, nonionizing approach for diagnosing and monitoring patients with PAH [30]. Numerous quantitative cMRI parameters have been documented to exhibit significant alterations in pulmonary hypertension, facilitating precise assessment of cardiac function and morphology. Notably, the emergence of vortical blood flow in the main pulmonary artery has been evaluated using 4D flow imaging. Furthermore, the length of vortical blood flow was proposed as a viable 4D flow measure for precise noninvasive assessment of PAH, making cMRI 4D flow imaging very compelling for noninvasive PAH monitoring [31]. Pulmonary artery hypertension using 4D flow can be seen in Figure 2.
3) Congenital heart disease: only information about tetralogy of Fallot is provided. I suggests including studies about patients in Fontan circulation or patients with transposition of the great arteries and suggest to reduce the part about tetralogy of Fallot.
Thank you for pointing this out. We agree with this comment. Therefore, we have updated the text:
Myocardial deformation analysis by cMRI-derived strain is an increasingly used method for evaluating ventricular function in patients receiving Fontan palliation. 2D cMRI feature tracking techniques that characterize RVGCS or RVGLS provide superior spatial resolution without constraints from acoustic windows. This presents a significant benefit for individuals with intricate ventricular morphology undergoing surgical palliation, leading to myocardial scarring and dyssynchrony. Research has more favoured strain-based measures to identify modest, subclinical alterations in ventricular contractile function [38]. Myocardial segment-specific strain-time curves characterize classic-pattern dyssynchrony and have recently been applied to patients after Fontan procedures [39]. In individuals with single right ventricle morphology, myocardial strain patterns undergo dynamic changes during and after single ventricle palliation. In individuals with hypoplastic left heart syndrome and its pathological variations, the RVGCS/RVGLS ratio may approach 1.0 and may completely revert in some patients long after the Fontan procedure [40].
Transposition of the great arteries after atrial switch surgery and congenitally corrected transposition of the great arteries are often linked to compromised systemic RV function and a poor prognosis. Diller et al. demonstrated that global longitudinal systolic strain is significantly diminished in patients with a systemic RV, correlates with subpulmonary ventricular function, and predicts adverse clinical outcomes in adults with atrial switch transposition of the great arteries [41]. Samarai et al. have shown that in patients with congenitally repaired transposition of the great arteries or those who have had Mustard or Senning operations, RVGLS may be beneficial for assessing systemic RV function [42].
4) I think that the manuscript would benefit from some general images about Feature tracking and the parameters that we get for RV and RA assessment.
Thank you very much for your review. Yes, we agree with you, but regrettably, at this time, we were unable to locate any images related to feature tracking or the parameters we use for RV and RA assessment.
5) In addition, a table that explains the measurements (e.g. atrial reservoir, conduit, and booster function) would also be helpful.
Thank you for pointing this out. We agree with this comment. Therefore, we have updated the text:
Table 2. The right atrium has three primary phases
|
Primary phases |
Physiological function
|
|
The reservoir phase
|
It functions as a storage for systemic blood following the closure of the tricuspid valve. This phase is contingent upon the RV's longitudinal contraction, the RA's compliance, and the caval venous blood flow return. An elevation during this phase contributes to enhanced RV filling [73].
|
|
The conduit phase |
It facilitates passive filling of the vena cava following the opening of the tricuspid valve. This phase relies on ageing and RV diastolic function [73].
|
|
The active contraction phase |
It is characharacterizedhe presence of sinus rhythm and functions as a "booster pump" for end-diastolic atrial contraction, thereby facilitating right RV filling. This phase relies on the contractile properties of the atrial wall and the ventricular end-diastolic pressure [73].
|
6) SENC / DENSE is better explained than routine FT but only feature tracking images are shown. Maybe this part in the text can be reduced.
Thank you for pointing this out. We agree with this comment. Therefore, we have updated the text:
Strain-encoded (SENC) cMRI, a recently developed tagged technology, outperformed the cMRI -FTI approach in terms of performance. The SENC technique incorporates through-plane tags to quantify strain in a direction that is perpendicular to the imaging plane [15]. Utilizing cMRI-based tissue tracking (cMRI-TT) allows for the retrospective investigation of dynamic deformation. Conversely, cMRI tagging can be utilized to investigate the process of ventricular untwisting [16].
Tissue displacement is converted into the image phase using displacement encoding with stimulated echoes (DENSE). DENSE quantitative cMRI detects myocardial strain and displacement with superior spatial and temporal sensitivities. A quick cine DENSE pulse sequence was established by echo combination reconstruction with intrinsic phase correction and balanced steady-state free precession (b-SSFP). This sequence produces quality strain with an effective breath-hold [17].
7) I recommend including a chapter about typical pitfalls (maybe before future directions).
Thank you for pointing this out. We agree with this comment. Therefore, we have updated the text:
- Typical pitfalls using strain imaging
The primary limitation of utilizing is attributed to inter-observer bias. Although numerous software companies have attempted to automate analysis, the majority remain only semi-automated and frequently necessitate manual adjustments to contours, thereby introducing the potential for human error. A study by the EACVI/ASE Industry Task Force reported a mean relative intra-observer difference in GLS of 6.1% and an inter-observer difference of 7.0%. Another significant limitation is the absolute difference in GLS among various vendors. The EACVI/ASE Industry Task Force undertook an Inter-Vendor Comparison, revealing a variance of 3% in the mean GLS across various vendor software platforms [74]. Exam times are also increasing as additional metrics are introduced to the procedure and 3D/4D becomes more widespread. The slow adoption of Strain can largely be attributed to its cost. Although many vendors provide strain modules tailored to their machines, the associated costs can be quite high, making obtaining strain analysis on more affordable or older equipment challenging. Also, accurate strain measurements rely heavily on training and experience [75].
Thank you very much for your review. We have uploaded the final version of the manuscript here.
Reviewer 2 Report
Comments and Suggestions for Authors
The authors examined the significance of CMR in cardiology, emphasizing its effectiveness in evaluating right ventricle size and function. They highlighted the necessity of early detection of subclinical cardiac changes to enhance treatment outcomes. This review explores the potential of strain imaging for offering prognostic insights into cardiovascular diseases, focusing on right ventricular strain and right atrial function, and aims to summarize current findings on right myocardial strain and biomechanics assessment using CMR.
Some suggestions:
I suggest integrating the prognostic aspects into the specific sections mentioned above to enhance readability. In addition to arrhythmogenic cardiomyopathy, the right ventricle plays a crucial role in various other cardiomyopathies, such as amyloidosis, hypertrophic cardiomyopathy, and dilated cardiomyopathy. Therefore, I recommend expanding this paragraph.
Finally, in the context of myocardial infarction, the right ventricle serves as an important prognostic parameter for predicting long-term outcomes, surpassing myocardial viability (see: DOI: 10.1111/echo.15854).
Lastly, I suggest adding a concluding paragraph that discusses the limitations of strain measurement, including vendor-specific limitations and other challenges.
Perform a thorough proofread to correct minor grammatical issues and ensure clarity and readability.
Comments on the Quality of English LanguageIt is recommended that abbreviations and typos be reviewed carefully.
Author Response
I suggest integrating the prognostic aspects into the specific sections mentioned above to enhance readability. In addition to arrhythmogenic cardiomyopathy, the right ventricle plays a crucial role in various other cardiomyopathies, such as amyloidosis, hypertrophic cardiomyopathy, and dilated cardiomyopathy. Therefore, I recommend expanding this paragraph.
Thank you for pointing this out. We agree with this comment. Therefore, we have updated the text:
We have integrated the prognostic chapter in all the specific sections.
5.5 Hypertrophic cardiomyopathy
cMRI-derived RV global strains and various regional longitudinal strains offer distinct radiological characteristics for differentiating between amyloidosis and HCM [52]. Right ventricular global strains correlate with outcomes in patients with HCM, including those without RV hypertrophy. RVGLS and RVGCS are independent predictors of HCM in cases with and without RV hypertrophy. Right ventricular global strains correlate with outcomes in patients with HCM, including those without RV hypertrophy. RVGLS and RVGCS serve as independent predictors of HCM in cases with and without RV hypertrophy [53]. Despite normal LVEF, Mahmod et al. demonstrated RV and LV strain impairment in HCM patients. RVEF decreased slightly but was within normal limits [54]. Although Mahmoud et al. found no association between RVGLS decline and cardiovascular composite events, it did predict non-sustained ventricular tachycardia. Hence, the independent predictive significance of the RV strain must be carefully examined with conventional cardiac function (54). RV long axis strain (RV-LAS) is the length change from the LV apex to the tricuspid annulus. Hence, it may include both ventricle's longitudinal functions. Yang et al. observed that RV-LAS predicted HCM patient's poor prognoses [55].
5.6. Amyloidosis
Right ventricular strain assessed via cMRI also serves as a significant prognostic marker for mortality in cardiac amyloidosis [56]. cMRI tissue tracking is a practical and reproducible RV deformation analysis method that could assist AL-type amyloidosis patients in detecting RV involvement early. AL-type amyloidosis mortality can be predicted by RVGLS [57]. Yang et al. demonstrated in a cMRI-FTI study on over 300 patients that mixing biventricular strain analysis alongside RVEF evaluation enhanced the detection of adverse outcomes: MACE predicted most accurately by RVGRS. cMRI-FTI deformation analysis of both ventricles has prognostic effectiveness comparable to traditional techniques [58].
5.7. Dilated cardiomyopathy
Recent studies indicate that RVGLS assessed through cMRI plays a significant role in reclassifying the risk of major cardiovascular events in patients with DCM [59]. The predictive value of the peak of RVGLS derived from cMRI-FTI in DCM patients with stage C or D heart failure (HF) and no atrial fibrillation (AF) has been documented by Liu et al. [60]. RV-LAS showcased the highest level of diagnosis accuracy in DCM patients, according to Arenja et al. Additionally, according to this research, RV-LAS was an independent MACE marker in multivariate analysis and essential in prognosis prediction [61].
Finally, in the context of myocardial infarction, the right ventricle serves as an important prognostic parameter for predicting long-term outcomes, surpassing myocardial viability (see: DOI: 10.1111/echo.15854).
Thank you for pointing this out. We agree with this comment. Therefore, we have updated the text:
In myocardial infarction, the right ventricle functions as a significant prognostic indicator for forecasting long-term outcomes, exceeding the importance of myocardial viability [51]
Lastly, I suggest adding a concluding paragraph that discusses the limitations of strain measurement, including vendor-specific limitations and other challenges.
Thank you for pointing this out. We agree with this comment. Therefore, we have updated the text:
- Typical pitfalls using strain imaging
The primary limitation of utilizing is attributed to inter-observer bias. Although numerous software companies have attempted to automate analysis, the majority remain only semi-automated and frequently necessitate manual adjustments to contours, thereby introducing the potential for human error. A study by the EACVI/ASE Industry Task Force reported a mean relative intra-observer difference in GLS of 6.1% and an inter-observer difference of 7.0%. Another significant limitation is the absolute difference in GLS among various vendors. The EACVI/ASE Industry Task Force undertook an Inter-Vendor Comparison, revealing a variance of 3% in the mean GLS across various vendor software platforms [74]. Exam times are also increasing as additional metrics are introduced to the procedure and 3D/4D becomes more widespread. The slow adoption of Strain can largely be attributed to its cost. Although many vendors provide strain modules tailored to their machines, the associated costs can be quite high, making obtaining strain analysis on more affordable or older equipment challenging. Also, accurate strain measurements rely heavily on training and experience [75].
Perform a thorough proofread to correct minor grammatical issues and ensure clarity and readability
Thank you very much for your review. Yes, we have checked our grammar and tried to make it as proficient as possible.
Thank you very much for your review. We have uploaded the final version of the manuscript here.
Reviewer 3 Report
Comments and Suggestions for Authors
The authors evaluated the role of cardiac magnetic resonance imaging in cardiology, emphasizing its reliability for assessing right ventricle size and performance. They highlighted the importance of identifying subclinical cardiac changes early in disease progression to improve treatment outcomes. The review discusses the potential of strain imaging to provide prognostic information for cardiovascular disease patients, focusing on right ventricle strain and right atrial function, and aims to summarize current data on right myocardial strain and biomechanics assessment using cardiac magnetic resonance imaging.
Issues/Suggestions:
In the abstract, it is important to specify that this is a narrative review, not a systematic one.
Since the title refers to the role of cardiac magnetic resonance imaging in assessing right ventricular function, particularly regarding right heart failure, it is necessary to expand the concepts beyond just right strain. In particular, it would be useful to evaluate the role of LGE in the RV, especially in the context of myocardial infarction, as well as the role of mapping techniques (e.g., hepatic mapping as a predictor of right ventricular dysfunction after STEMI; cite: PMID: 39364943). I therefore suggest expanding the review to include this information.
A careful review of the references is essential.
I finally recommend including a summary table with the parameters of right ventricular function derived from cardiac magnetic resonance imaging, detailing the clinical contexts in which they have proven useful. Such a table would be very beneficial for readers to summarize the main findings.
Comments on the Quality of English LanguageModerate English revision is recommended.
Author Response
In the abstract, it is important to specify that this is a narrative review, not a systematic one.
Thank you very much for your review. Yes, I did specify it in the abstract and also in the main text.
Since the title refers to the role of cardiac magnetic resonance imaging in assessing right ventricular function, particularly regarding right heart failure, it is necessary to expand the concepts beyond just right strain. In particular, it would be useful to evaluate the role of LGE in the RV, especially in the context of myocardial infarction, as well as the role of mapping techniques (e.g., hepatic mapping as a predictor of right ventricular dysfunction after STEMI; cite: PMID: 39364943). I therefore suggest expanding the review to include this information.
Thank you for pointing this out. We agree with this comment. Therefore, we have updated the text:
The identification of the pattern of LGE presentation is essential. LGE in the ischaemic pattern consistently affects the subendocardial layer, exhibiting varying degrees of transmural involvement. It is confluent and localized within the territory of a single coronary artery. In a non-ischaemic pattern, LGE does not meet the established criteria, presenting as midwall, subepicardial, or mixed, and is not necessarily confluent or restricted to the territory of a single coronary artery. Using the secondary eigenvector from cardiac diffusion tensor imaging, the assessment of sheetlet orientation changes correlates with impaired radial strain. Segments exhibiting reduced subendocardial cardiomyocytes, indicated by a diminished proportion of myocytes displaying right-handed orientation on helix angle maps, demonstrate compromised longitudinal strain. Infarct segment enhancement shows a significant correlation with the secondary eigenvector and right-handed orientation. The data indicate a correlation between myocardial microstructure and contractility post-myocardial infarction, implying a possible clinical significance of cMRI cardiac diffusion tensor imaging for functional outcomes [48].
Furthermore, hepatic T1 mapping has emerged as a potential imaging biomarker for the cardio-hepatic axis in ST-elevation myocardial infarction, correlating with right ventricular involvement and elevated NT-proBNP levels [49].
A careful review of the references is essential.
Thank you very much for your review. Yes, we have checked our references and tried to make it as proficient as possible.
I finally recommend including a summary table with the parameters of right ventricular function derived from cardiac magnetic resonance imaging, detailing the clinical contexts in which they have proven useful. Such a table would be very beneficial for readers to summarize the main findings
- Conclusions
Table 4 summarizes the parameters of right ventricular function derived from cardiac magnetic resonance imaging and details the clinical contexts in which they have proven useful.
Table 4: Parameters of right ventricular function derived from cardiac magnetic resonance imaging and their clinical function
|
Cardiac magnetic resonance right ventricular parameter |
Clinical function |
|
RVGLS |
Independent prognostic factor for adverse outcomes in DCM, HCM, ARVC, and AL type amyloidosis [10]. RVGLS predicts MACE independently of clinical and imaging factors in PAH patients [28]. TOF patients had significantly reduced RVGLS, and it was strongly connected with poor outcomes. RVGLS predicted adverse arrhythmic outcomes in TOF patients [35]. RVGLS may be beneficial for assessing systemic RV function in transposition of the great arteries or those who have had Mustard or Senning operations [42]. Patients with ARVC who have sustained VA exhibit a decrease in RVGLS [45]. The RVGLS demonstrated an independent prediction value for functional status in patients with RV infarction [50].
|
|
RVGRS |
The RVGRS of patients with PAH presented a statistically significant decrease compared to healthy individuals [27]. MACE is predicted most accurately by RVGRS in cardiac amyloidosis [57].
|
|
RVGCS |
The RVGCS of patients with PAH presented a statistically significant decrease compared to healthy individuals [27]. Patients with ARVC who have sustained VA exhibit a decrease in RVGCS [45].
|
|
RVFWLS |
RVFWLS parameter exhibited the capability to differentiate between acute ischemia and non-ischemic myocardium in patients with RV infarction [50]. |
|
Right atrial strain |
pPAH patients had impaired RA strain versus healthy controls [28]. RA changed even in pPAH with maintained RVEF. This group of pPH patients had reduced conduit strain and RA reservoir [28]. All RA phasic strain factors independently predicted MACE in PAH patients [28]. Reduced RAGLS indicates lower RA reservoir function in TOF patients [28]. RA reservoir strain and conduit strain independently predicted all-cause mortality in DCM patients [60]. RA strain outperformed baseline clinical and cMRI indicators in predicting MACE in DCM patients [65]. |
Abbreviations: ARVC, Arrhythmogenic right ventricular cardiomyopathy; DCM, dilated cardiomyopathy; HCM, hypertrophic cardiomyopathy; RA, right atrium; RVGCS, RV global circumferential strain; RVGLS RV global longitudinal strain; RVGRS, RV global radial strain; RVFWLS, RV free wall longitudinal strain RVFWLS; RV-LAS, the length change from the LV apex to the tricuspid annulus; RAGLS, RA global longitudinal strain; RVEF, RV ejection fraction; pPAH, precapillary PAH, pulmonary arterial hypertension; MACE, major adverse cardiac events; TOF, tetralogy of Fallot; VA, ventricular arrhythmias;
Thank you very much for your review. We have uploaded the final version of the manuscript here.
Round 2
Reviewer 2 Report
Comments and Suggestions for Authors
The authors have addressed my previous comments, and I have no further questions.
Reviewer 3 Report
Comments and Suggestions for Authors
Congratulations to the authors, as the manuscript has improved after the revision.